# Leptin-induced Trafficking of K_ATP_ Channels: A Mechanism to Regulate Pancreatic β-cell Excitability and Insulin Secretion

**DOI:** 10.3390/ijms20112660

**Published:** 2019-05-30

**Authors:** Veronica Cochrane, Show-Ling Shyng

**Affiliations:** Department of Chemical Physiology & Biochemistry, Oregon Health & Science University, Portland, OR 97239, USA; shyngs@ohsu.edu

**Keywords:** Leptin, ATP-sensitive potassium (K_ATP_) channel, β-cell, Insulin secretion

## Abstract

The adipocyte hormone leptin was first recognized for its actions in the central nervous system to regulate energy homeostasis but has since been shown to have direct actions on peripheral tissues. In pancreatic β-cells leptin suppresses insulin secretion by increasing K_ATP_ channel conductance, which causes membrane hyperpolarization and renders β-cells electrically silent. However, the mechanism by which leptin increases K_ATP_ channel conductance had remained unresolved for many years following the initial observation. Recent studies have revealed that leptin increases surface abundance of K_ATP_ channels by promoting channel trafficking to the β-cell membrane. Thus, K_ATP_ channel trafficking regulation has emerged as a mechanism by which leptin increases K_ATP_ channel conductance to regulate β-cell electrical activity and insulin secretion. This review will discuss the leptin signaling pathway that underlies K_ATP_ channel trafficking regulation in β-cells.

## 1. Introduction

Leptin is a 167 amino acid peptide hormone belonging to the cytokine family that was originally identified as a ‘satiety hormone’ due to its central actions in the hypothalamus as an appetite suppressant. The subsequent finding that leptin is predominantly produced and secreted by white adipocytes, and plasma leptin levels are highly correlated with body fat mass, further supported the notion that leptin was the long sought-after hormone responsible for coordinating body weight with food intake and energy expenditure to regulate energy homeostasis. Although the role of leptin in energy homeostasis has garnered much attention, leptin has also been implicated in numerous other biological processes including glucose homeostasis.

Evidence that leptin was critical for regulating glucose homeostasis first came from mice deficient in leptin, *ob/ob* mice. In addition to developing obesity, indicative of an energy imbalance, *ob/ob* mice exhibit hyperinsulinemia, insulin resistance, and high blood glucose levels recapitulating type 2 diabetes in humans [1,2]. It is important to note that *ob/ob* mice develop hyperinsulinemia and hypoglycemia prior to developing insulin resistance, obesity and hyperglycemia. Moreover, administering leptin to *ob/ob* mice normalizes serum insulin and glucose concentrations before any changes in the animals’ weight are observed [3]. From these findings it became evident that leptin serves a direct role in maintaining glucose homeostasis.

Pancreatic β-cells are the sole source of insulin for the body and are indispensable to glucose homeostasis. β-cells are stimulated to secrete insulin by elevated levels of blood glucose. Insulin then signals to target tissues such as muscle, liver and fat to uptake glucose for usage or storage thereby restoring resting blood glucose levels [4]. Although many of the effects of leptin on glucose homeostasis are attributed to actions of leptin in the central nervous system there is growing evidence that the peripheral actions of leptin are important as well. In light of the finding that leptin rapidly improves serum insulin levels in *ob/ob* mice pancreatic β-cells were identified as a potential peripheral target of leptin. Indeed, several groups have since shown that leptin receptors are expressed on β-cells and that leptin reduces insulin secretion when directly applied to isolated islets [5,6,7,8,9,10,11]. Insulin, in addition to promoting glucose uptake, stimulates adipogenesis as well as the production and secretion of the adipocyte hormone leptin. Together, these observations led to the proposal that leptin and insulin function in a dual hormonal feedback loop termed the “adipoinsular axis” to maintain fat mass and blood glucose levels within the physiological range [12]. Disruption of this signaling axis as a result of insulin or leptin resistance is expected to upset the balance of fat and glucose regulation and contribute to the pathology of obesity and diabetes. This sets the stage for the ensuing studies in the past two decades to understand the mechanisms by which leptin regulates insulin secretion in β-cells and the physiological role of this regulation.

In pancreatic β-cells, a key link between serum glucose levels and insulin secretion is the ATP-sensitive potassium (K_ATP_) channel which sets the β-cell resting membrane potential. K_ATP_ channels are regulated by the intracellular ATP/ADP ratio, enabling them to effectively serve as metabolic sensors that couple serum glucose to insulin secretion [13,14,15]. At low blood glucose concentrations, the low intracellular ATP/ADP ratio favors K_ATP_ channel opening, which hyperpolarizes the cell and thus inhibits insulin secretion. Conversely, at high glucose concentrations, the increased intracellular ATP/ADP ratio favors K_ATP_ channel closure, resulting in cell depolarization, activation of voltage-gated calcium channels, and insulin exocytosis. Interestingly, at high glucose concentrations when the majority of K_ATP_ channels are mostly closed and β-cells are depolarized leptin was found to cause β-cell hyperpolarization by increasing K_ATP_ channel conductance. Although this provides a mechanism by which leptin suppresses insulin secretion it was not known how leptin increased K_ATP_ channel conductance.

Modulating channel gating or the abundance of channels in the cell membrane can alter total cellular K_ATP_ channel conductance. Compared to gating regulation, little was known about how regulation of K_ATP_ channel surface density may contribute to β-cell function. Recent studies found that leptin does not alter K_ATP_ channel gating, but instead promotes channel trafficking to the β-cell surface. The increased surface density of K_ATP_ channels in response to leptin increases K_ATP_ conductance and pushes the cell towards a hyperpolarized state, which is expected to suppress insulin secretion. Therefore, leptin-mediated K_ATP_ channel trafficking has emerged as a novel mechanism for regulating β-cell electrical activity and insulin secretion. It is worth noting that leptin also inhibits insulin secretion via K_ATP_-independent pathways [11,16,17]. However, this review will focus on the advances that have been made to delineate the mechanism by which leptin promotes K_ATP_ channel translocation in β-cells.

## 2. Leptin Increases K_ATP_ Channel Conductance in β-Cells

The discovery that leptin directly inhibits insulin secretion from islets ex vivo prompted the question as to how this occurs. K_ATP_ channels were an obvious candidate since their activity was already known to regulate the resting potential of β-cells, hence insulin secretion. Indeed, leptin was found to cause membrane hyperpolarization by increasing β-cell membrane K^+^ conductance, which could be reversed upon application of a K_ATP_ channel specific sulfonylurea inhibitor, tolbutamide [6,18]. In these studies, leptin was found to induce hyperpolarization at doses from 1–10 nM, which are within the physiological range [19,20]. These findings suggested that leptin reduced insulin secretion by enhancing K_ATP_ channel activity, but the underlying mechanism remained elusive.

Several follow-up studies by the Ashford lab attempted to elucidate how leptin signaling enhanced K_ATP_ channel activity. This work was conducted under the assumption that leptin increased activity of K_ATP_ channels already present in the membrane. Their initial studies focused on testing the ability of leptin to increase phosphatidylinositol (3,4,5)-trisphosphate (PIP_3_) levels by activating phosphoinositide 3-kinase (PI3K), as PIP_3_ has been shown to increase K_ATP_ channel activity. Although application of exogenous PIP_3_ did increase K_ATP_ channel activity [21], it was revealed that the increase of endogenous PIP_3_ levels in response to leptin stimulation was insufficient [22,23]. Moreover, they found that PIP_3_ levels increased as the result of phosphatase and tensin homolog (PTEN) inhibition rather than PI3K activation. PTEN is a dual lipid and protein phosphatase that has been extensively studied for its role in tumorigenesis with the majority of its actions being attributed to its phosphatase activity towards lipids especially PIP_3_ [24]. In addition to increasing PIP_3_ levels PTEN inactivation by leptin was shown to cause actin remodeling. This was a particularly interesting finding since a prior study in cardiomyocytes had shown that actin destabilizing agents could increase K_ATP_ channel activity [25]. It was further determined that the effect leptin has on β-cell K_ATP_ channels is dependent on actin depolymerization and could be recapitulated by actin destabilizing agents [22,23,26]. Expression of PTEN mutants revealed that inhibition of both PTEN lipid and phosphatase activities are required for leptin signaling. In the sections below we will revisit the role of PTEN in the context of K_ATP_ channel trafficking and discuss potential downstream effectors involved in actin remodeling.

## 3. Leptin Increases K_ATP_ Channel Surface Density

In recent years, increased K_ATP_ channel surface density has emerged as the predominant mechanism by which leptin increases K_ATP_ channel conductance. This finding was first reported by Park et al. and months later by our group [27,28]. These studies were primarily conducted using 10 nM leptin at 11mM glucose (see sections below). Our lab found that following leptin treatment K_ATP_ channels do not display altered sensitivity to ATP or ADP indicating that the effects of leptin are not due to changes in K_ATP_ channel gating [28]. Using surface biotinylation assays to monitor K_ATP_ channel surface density in the rat insulinoma cell line INS-1 clone 832/13 (INS-1) [29], there was a detectable increase observed as early as 5 min following leptin treatment with the response peaking at 30 min and signs of reversal within 60 min [27,28]. The transient nature of the response is likely to be physiologically significant and may serve to prevent prolonged inhibition of insulin secretion. The increase in K_ATP_ channel surface density was primarily due to insertion of K_ATP_ channels into the β-cell membrane and to a lesser extent reduced K_ATP_ endocytosis. Based on the time frame in which K_ATP_ channel surface density increases, it is likely that the channels reside in a readily available vesicle reserve rather than increased K_ATP_ channel biogenesis. Consistent with this, there was no apparent change in total K_ATP_ channel protein content over the course of leptin treatment. Of note, the increase of surface K_ATP_ channels within 5 min coincides with leptin-induced K_ATP_ channel conductance and hyperpolarization [6,18,30]. By analyzing K_ATP_ channel currents it is estimated that ~800 channels are inserted into the β-cell membrane in response to leptin and that the total number of K_ATP_ channels in the membrane increases from ~400 to ~1200, which is in agreement with the 3- to 4-fold increase observed using surface biotinylation [27,28]. Importantly, blocking the ability of leptin to increase K_ATP_ channel surface density by blocking downstream signaling events (see below) also prevented cell hyperpolarization, confirming K_ATP_ channel surface density regulation as a mechanism for modulating β-cell excitability.

Although the majority of the mechanistic studies have been conducted using INS-1 cells there is substantial evidence that this mechanism is conserved in rodent and human β-cells. Park et al. confirmed the effects of leptin on K_ATP_ channels in isolated mouse β-cells via immunostaining and whole-cell currents [27]. Our group has demonstrated that isolated human β-cells also hyperpolarize in response to leptin and as will be discussed in the following sections, inhibiting leptin signaling molecules necessary for K_ATP_ channel trafficking blocks human β-cell hyperpolarization [30,31]. Collectively, the above findings provided the foundation for establishing K_ATP_ channel trafficking as a physiologically relevant mechanism by which leptin inhibits insulin secretion.

## 4. K_ATP_ Channel Trafficking Signaling Molecules

Since the initial reports that leptin regulates K_ATP_ channel trafficking, there has been significant progress towards delineating the signaling cascade underlying this regulation. A model based on published studies to date is shown in Figure 1. In this model, the activation of leptin receptors by leptin leads to an influx of calcium (Ca^2+^) that stimulates calcium/calmodulin-dependent kinase kinase β (CaMKKβ) to phosphorylate and activate AMP-activated protein kinase (AMPK). Downstream of AMPK actin remodeling occurs to allow trafficking of vesicles containing K_ATP_ channels to the β-cell surface. The following sections will evaluate the role of the signaling molecules that have been implicated in leptin mediated K_ATP_ channel trafficking.

### 4.1. AMPK

AMPK is known for its role as a master regulator of cellular energy and glucose homeostasis [32]. Activation of AMPK in response to high cellular AMP or ADP levels promotes catabolic processes to increase ATP concentrations and inhibits anabolic processes. A role of AMPK in K_ATP_ channel trafficking was first documented by Lim et al. who showed that under low glucose culturing conditions (0–3 mM), K_ATP_ channels translocate to the β-cell membrane in an AMPK-dependent manner to reduce insulin secretion [33]. This study uncovered a physiological mechanism by which β-cells adapt to glucose starvation. Interestingly, a subsequent study by Beall et al. showed that β-cells lacking AMPKα2 or expressing a kinase-dead AMPKα2 failed to hyperpolarize under low glucose conditions [34]. Although K_ATP_ channel trafficking was not examined, dysregulation of K_ATP_ trafficking likely contributed to the observed β-cell dysfunction.

In addition to responding to changes in intracellular energy levels AMPK activity is modulated by various hormones including leptin. The effect of leptin on AMPK activity is cell-type specific. For example, in skeletal muscle leptin stimulates AMPK [35] whereas it inhibits AMPK in parts of the hypothalamus [36]. In the case of INS-1 and mouse β-cells, leptin treatment was found to cause a marked increase in phosphorylation of the AMPK catalytic α subunit at Threonine 172 (pAMPKα) and its substrate Acetyl-CoA carboxylase (pACC) indicative of increased AMPK activity [27,28]. The increase in pAMPKα was transient and peaked at 30 min similar to the changes observed in surface K_ATP_ channel density. To test whether AMPK is involved in leptin-induced K_ATP_ channel trafficking both pharmacological and genetic approaches were employed. While the inhibition of AMPK occluded the effects of leptin, AMPK activation mimicked the effects of leptin on K_ATP_ channel translocation [27,28]. These findings provide strong evidence that leptin activates AMPK to recruit K_ATP_ channels to the β-cell surface.

### 4.2. CaMKKβ

AMPK may be activated in an AMP-dependent or Ca^2+^-dependent manner [32]. AMP binding causes allosteric activation, but it also promotes phosphorylation of the AMPKα subunit by LKB1, which greatly enhances AMPK activation (>100 fold). Alternatively, AMPK may be activated by a rise in cytoplasmic calcium levels activating CaMKKβ, which then phosphorylates AMPKα. Thus, while the observed increase in AMPK activity in response to glucose starvation [33] most probably occurs in an AMP-dependent manner this is unlikely to be the case at higher glucose conditions.

CaMKKβ belongs to the family of Ca^2+^/CaM kinases and as such CaMKKβ is activated when bound to Ca^2+^/CaM. CaMKKβ signaling through CaM kinases or AMPK is important for a myriad of physiological processes including glucose homeostasis [37]. To determine whether leptin signals through CaMKKβ to activate AMPK, CaMKKβ was knocked down by siRNA or pharmacologically inhibited. Inhibition of CaMKKβ completely ablated leptin activation of AMPK whereas LKB1 knockdown had no significant effect [27,31]. Furthermore, an increased K_ATP_ channel surface density in response to leptin was also dependent on CaMKKβ activity. From these studies it was concluded that leptin signals via CaMKKβ to activate AMPK. Interestingly, CaMKKβ has been identified as a regulator of insulin secretion; islets from CaMKKβ knockout mice showed increased insulin secretion [38]. While the mechanism remains to be elucidated, the role of CaMKKβ in activating AMPK to stimulate K_ATP_ channel trafficking is likely to be a factor.

### 4.3. PI3 Kinase and TRPC4 Channels

Taking into consideration that CaMKKβ activation is dependent on Ca^2+^ it follows that leptin must somehow elevate intracellular calcium levels. Consistently, Ca^2+^-chelating agents such as BAPTA-AM and EGTA completely blocked leptin activation of AMPK and leptin-induced increase of K_ATP_ surface expression [27,31]. To identify where Ca^2+^ is coming from, Park et al. used pharmacological inhibitors and ruled out voltage-gated Ca^2+^ channels or the ER Ca^2+^ store as the source [27]. They focused on transient receptor potential channels (TRPC) instead based on attenuation of the leptin effect by the drug 2-aminoethyldiphenyl borate (2-APB). Here, it is worth noting that although 2-APB does block TRPC channels, it also blocks a number of other channels including InsP3-receptors and store-operated Ca^2+^ channels. Nevertheless, they went on to show that knocking down TRPC4 but not TRPC5 by siRNA or inhibition of TRPC4 by ML-204 abrogated the effect of leptin. Moreover, they concluded that leptin activates TRPC4 channels by activating PI3-kinase because LY294002, a PI3K inhibitor, abolished leptin-induced increase of TRPC4 conductance. Of note, an early study by Harvey et al. also implicated a role of PI3K in leptin regulation of K_ATP_ channel activity, again using PI3K inhibitors [21]. However, it is important to keep in mind that it is now known that LY294002 also directly inhibits some potassium channels [39], thus results should be interpreted with caution.

Although TRPC4 channels were identified as the calcium source for leptin by Park et al [27], we were unable to block the leptin effect using the TRPC4 inhibitor ML-204 [30]. Instead, we have identified NMDA receptors as the mediator for Ca^2+^ entry in response to leptin [30]. The reason for this discrepancy remains unclear but may depend on experimental conditions. Below, we will discuss the evidence for the *N*-Methyl-d-Aspartate (NMDA) receptor as a calcium source for leptin signaling.

### 4.4. NMDA Receptors

NMDA receptors (NMDARs) are ionotropic glutamate receptors that are permeant to monovalent cations and Ca^2+^ [40]. They are activated by co-agonists glutamate and glycine and require membrane depolarization to remove external Mg^2+^ block for conduction. Upon activation, NMDARs allow an influx of calcium into cells and trigger various calcium-dependent signaling cascades. NMDARs are highly expressed throughout the central nervous system where they play important roles in synaptic plasticity. Interestingly, a prior study had linked leptin to enhanced NMDAR activity for long-term potentiation in hippocampal neurons [41]. Although NMDARs have been primarily studied in the brain there is increasing evidence that they have important function in pancreatic β-cells [42]. A recent study showed that NMDAR antagonism increased glucose-stimulated insulin secretion (GSIS) both in vivo and in vitro; moreover, this effect was dependent on the expression of functional K_ATP_ channels in islets [43].

Our group investigated whether NMDARs play a role in recruitment of K_ATP_ channels in β-cells. We found that direct activation of NMDARs increased K_ATP_ channel surface density and induced β-cell hyperpolarization similar to leptin [30]. It is noteworthy that this was specific to NMDARs as targeting another ionotropic glutamate receptor, the AMPA type receptor, had no effect. Since NMDARs had previously been shown to activate CaMKKβ-AMPK signaling in hippocampal neurons it was plausible that NMDARs and leptin share a common signaling cascade [44]. In support of this, the effect of NMDAR activation on K_ATP_ channel trafficking is also dependent on CaMKKβ and AMPK activity [30].

Calcium imaging experiments provided direct evidence that leptin induces calcium influx through NMDARs. This was observed using a high affinity calcium sensor (Fluo-4) in the absence of glucose in order to detect small changes in intracellular calcium without interference from glucose-triggered calcium events. NMDAR antagonists abolished leptin-induced Ca^2+^ signals, while NMDA agonists induced Ca^2+^ influx in the absence of leptin [30]. We note our finding that leptin increases intracellular Ca^2+^ concentrations contradicts earlier studies [6,11] likely because of improved Ca^2+^ imaging reagents and methods. Importantly, we showed that leptin potentiation of NMDARs and the role of NMDARs in K_ATP_ channel trafficking is conserved between INS-1 cells and human β-cells [30]. These studies have identified a novel role for NMDARs in β-cells in mediating leptin signaling and provide a potential mechanism as to how inhibition of NMDARs enhances insulin secretion.

The discovery that activation of NMDARs promotes K_ATP_ channel trafficking in the absence of leptin also raises interesting questions about their functional role aside from mediating the effect of leptin. As was previously mentioned, glutamate is an essential endogenous ligand for NMDA receptors. Potential sources of glutamate in islets include innervating neurons, β-cells as well as α-cells [42]. Therefore, physiological cues that result in these sources releasing glutamate may serve to modulate insulin secretion by activating NMDA receptors. In this regard, it is interesting to note that glutamate has been shown to be co-released with glucagon by α-cells. Although it is proposed that the co-released glutamate serves as a positive feedback autocrine signal for glucagon secretion by activating AMPA/Kainate subtype glutamate receptors [45], an interesting possibility is that glutamate may also serve as a paracrine signal to inhibit insulin secretion by increasing K_ATP_ channel trafficking to the β-cell surface.

Collectively, the above studies detail a previously unknown signaling mechanism involving NMDAR, CaMKKβ, and AMPK by which leptin promotes trafficking of K_ATP_ channels. This mechanism may help to explain previous studies that have implicated these molecules in the regulation of insulin secretion. However, as discussed below the steps downstream of AMPK required for actin remodeling remain poorly understood.

### 4.5. Actin Remodeling Molecules

Although early studies presumed that leptin-induced actin depolymerization activated K_ATP_ channels already present in the β-cell membrane, more recent studies have concluded that actin remodeling allows trafficking of vesicles containing K_ATP_ channels to the cell surface. Cortical actin is predicted to serve as a barrier for vesicle exocytosis, and as such actin remodeling typically occurs prior to exocytotic events [46]. It has been shown that actin remodeling downstream of AMPK in the leptin signaling cascade is a necessary step for K_ATP_ channel trafficking to the membrane [28]. AMPK has previously been shown to be involved in actin depolymerization in other cell types but how it does so in β-cells is still unresolved. Below we will discuss potential signaling molecules that may act downstream of AMPK.

#### 4.5.1. PKA

PKA is known to modulate the activity of many molecules involved in actin dynamics and has been shown to regulate the trafficking of a number of ion channels [47,48,49]. Studies by our lab found that PKA is involved in actin remodeling downstream of AMPK [28]. Activation of PKA directly by cAMP elevating agents mimics the effects of leptin and an AMPK activator, while pharmacological inhibition of PKA blunts leptin or AMPK activator-induced actin depolymerization and K_ATP_ channel trafficking. These findings raise the possibility that AMPK may activate PKA to cause actin depolymerization necessary for channel trafficking. Interestingly, crosstalk between AMPK and PKA has been reported in smooth muscle cells and cardiomyocytes where activation of AMPK results in increased PKA activity [50,51]. Although there is no established mechanism for how PKA may be activated by AMPK, AMPK has been shown to modulate the activity of phosphodiesterases [52], which could potentially increase cAMP levels and activate PKA. However, in the absence of a mechanism by which AMPK activates PKA, a permissive role of PKA in allowing leptin-induced actin remodeling rather than being part of the signaling cascade cannot be ruled out.

Many PKA effector proteins involved in actin dynamics have been identified including LimK, Cofilin, and Rac [53,54]. Interestingly, leptin has recently been shown to increase Rac activity in INS-1 cells, but it has yet to be determined whether this is dependent on PKA (see below).

#### 4.5.2. PTEN

Soon after leptin was found to recruit K_ATP_ channels to the β-cell surface the role of PTEN, which was previously implicated in mediating the effect of leptin on K_ATP_ conductance, was re-examined. In agreement with early studies, inhibition of PTEN’s lipid and phosphatase activities are required for actin remodeling and leptin-induced K_ATP_ channel trafficking [55,56]. However, in contrast to findings by the Ashford lab [23], PTEN inactivation was shown to require phosphorylation by Glycogen Synthase Kinase 3 β (GSK3β) but not Casein Kinase 2 (CK2) [55]. It was determined that GSK3β lies downstream of AMPK, but the relationship between the two is unclear as GSK3β activation requires phosphorylation of Tyrosine 216 and AMPK is a Ser/Thr kinase [55].

In an effort to elucidate how PTEN inactivation following leptin stimulation leads to actin remodeling, Han et al. examined the role of Rac [56], a Rho family small GTPase molecular switch that is known to regulate actin dynamics for vesicle transport [57]. They showed that Rac activity is increased and that this is necessary for the recruitment of K_ATP_ channels to the β-cell surface [56]. As PIP_3_ has been reported to enhance Rac activity by recruiting guanine nucleotide exchange factors (GEFs) [58,59,60], PIP_3_ elevation following PTEN inactivation could be an underlying mechanism of actin depolymerization. Following Rac activation, there was evidence for increased activation of Myosin II [56], which has also been implicated in actin disassembly and vesicular trafficking [61]. Thus, these studies proposed that leptin signaling inhibits PTEN to activate Rac and Myosin II, leading to actin depolymerization and trafficking of K_ATP_ channels [56].

Although the above studies provide a framework for understanding how PTEN participates in the leptin signaling pathway to regulate K_ATP_ channel trafficking, many details are still missing. Also, studies involving actin and actin regulatory proteins are not always straightforward. Many of these proteins are also modulated by PKA or AMPK directly or indirectly and global cellular manipulation of these molecules can have unintended consequences that confound data interpretation. Moving forward it will be important to design more sophisticated methods and molecular tools to evaluate actin remodeling events specific to K_ATP_ channel trafficking regulation by leptin.

## 5. The Role of Glucose and its Interplay with Leptin in K_ATP_ Channel Trafficking Regulation

Before leptin was shown to regulate K_ATP_ channel trafficking, glucose had already been reported to modulate K_ATP_ trafficking. Yang et al. first showed that high glucose promotes recruitment of K_ATP_ channels to the β-cell membrane in a Ca^2+^ and PKA-dependent manner [62]. Subsequently, Lim et al. showed that glucose starvation mobilizes K_ATP_ channel to the β-cell surface by activating AMPK [33]. Conceptually, increased K_ATP_ trafficking to the β-cell surface is a proper physiological response to low glucose as it is expected to reduce insulin secretion, but why would β-cells recruit more K_ATP_ channels to the membrane at high glucose? Yang et al. proposed that the increased K_ATP_ surface density at high glucose concentrations may be an autoinhibitory feedback mechanism to prevent over secretion of insulin and prime the cell for subsequent glucose-stimulated insulin exocytosis. However, complicating the matter, a recent study found high glucose reduces, rather than increases, K_ATP_ channel surface density by stimulating channel endocytosis and proposed that this is the primary mechanism by which glucose depolarizes β-cell membrane potential and stimulates insulin secretion [63]. While intriguing, the notion that K_ATP_ channel endocytosis is the primary mechanism driving β-cell depolarization in response to glucose needs to be rigorously examined. Numerous neonatal diabetes-causing mutations in K_ATP_ channels have been shown to markedly reduce surface expression and yet patients fail to secrete insulin because these mutations alter channel gating response to intracellular ATP and ADP [64,65,66]. Moreover, there is an overwhelming number of congenital hyperinsulinism mutations that do not affect channel expression but stimulate insulin secretion at low glucose due to defective gating response [67,68]. Clearly, more work is needed to resolve the apparent conflicting findings from different groups.

Given that both glucose and leptin affect K_ATP_ channel trafficking, an important question is how the two signaling pathways relate to each other to impact insulin secretion. Park et al. addressed this issue by examining β-cells from either WT or *ob/ob* mice lacking leptin under fed or fasted conditions [27]. They found that β-cells in WT islets showed more cell periphery K_ATP_ channel staining and increased K_ATP_ conductance in the fasted state (blood glucose ~7.7 mM) compared to the fed state (blood glucose ~13.5 mM), suggesting that K_ATP_ channel translocation is responsive to feeding status in vivo. Interestingly, in β-cells from *ob/ob* islets there was no evidence of K_ATP_ channel translocation or increase in conductance in the fasted state compared to the fed state; however, leptin injection recovered channel responses to fasting glucose. These observations suggest that leptin is needed for the regulation of K_ATP_ channel density in vivo in response to blood glucose fluctuations. Of note, in the initial study, which demonstrated activation of AMPK and increased K_ATP_ channel surface density following glucose deprivation in INS-1 cells, prolonged (2 h) incubation in non-physiologically low glucose of less than 3 mM was needed [33]. It is therefore proposed that leptin functions to elevate AMPK activity sufficiently under physiologically relevant fasting glucose concentrations. This study implicates the importance of leptin in tuning the sensitivity of AMPK to trigger a physiological response to falling glucose. Furthermore, it suggests that the leptin-insulin feedback mechanism may be important for preventing blood glucose from falling to a dangerously low level.

## 6. Future Questions

Although great efforts have been made to elucidate the signaling mechanism by which leptin suppresses insulin secretion, there are still many outstanding questions. For instance, it has not been established how leptin leads to potentiation of NMDA receptors. Studies in hippocampal neurons have implicated a role of Src kinase [41] in augmenting NMDAR activity. Whether this is applicable to leptin signaling in β-cells remains to be seen.

Many of the molecules involved in leptin signaling (Ca^2+^, cAMP, PKA, and Rac) have also been identified as positive regulators of insulin secretion. For example, cAMP elevating agents have been found to both increase K_ATP_ channel surface density and enhance insulin secretion. A scenario that could reconcile these paradoxical findings is that there are distinct cAMP signaling microdomains created by subcellular organization of specific signaling molecules within β-cells involved in different processes. Understanding the spatiotemporal regulation of leptin signaling would aid future investigations into how β-cells coordinate and integrate signaling events to achieve a proper physiological response.

As noted earlier, the effect of leptin on K_ATP_ channel surface density is transient. There have been no studies conducted to investigate the transient nature of the leptin response. Although there are several possible steps at which leptin signaling may be shut off, they remain untested. Identifying pertinent negative feedback mechanisms could help us to gain insight into how leptin resistance may occur at the β-cell level in pathological states such as obesity or type 2 diabetes.

In addition to K_ATP_ channels we have shown in the rat β-cell line INS-1 that leptin also increases surface density of the voltage-gated delayed rectifier channel Kv2.1 but not several other ion channels including G-protein coupled Kir channels (GIRK), sodium and calcium currents [31]. Remarkably, trafficking of Kv2.1 and K_ATP_ channels in response to leptin occurs through a common mechanism involving NMDAR, CaMKKβ, AMPK, and PKA. As Kv2.1 channels are the dominant delayed inward rectifier and have a prominent role in the repolarization of action potentials in β-cells, at least in mice [69], the concerted regulation of K_ATP_ and Kv2.1 channels by leptin would synergistically reduce β-cell excitability at elevated glucose to suppress insulin secretion. The finding that K_ATP_ and Kv2.1 channels are co-regulated by leptin raises several questions. Is Kv2.1 trafficking also regulated by glucose? Do K_ATP_ and Kv2.1 channels reside in the same intracellular vesicles that traffic to the cell surface in response to leptin? There is currently no consensus on the intracellular vesicle population in which K_ATP_ channels reside. Insulin granules, chromogranin positive but insulin negative dense core granules, as well as EEA1 positive early endosomes have all been reported to contain K_ATP_ channels [27,62,70,71], while few studies have directly examined the subcellular localization of Kv2.1 channels. Furthermore, are there yet unidentified channels or transporters that are regulated by leptin in β-cells such as AMPARs, which have been shown to be regulated by leptin in hippocampal neurons?

Finally, what is the contribution of direct leptin signaling on β-cells in glucose homeostasis? Despite clear evidence that leptin regulates K_ATP_ and Kv2.1 channel trafficking to suppress β-cell excitability, a definitive demonstration that leptin’s action on β-cells has a role in regulating blood glucose is, surprisingly, still lacking. Most signaling mechanism studies have been conducted using β-cell lines or primary β-cells with biochemical or electrophysiological readouts as endpoints. It would be important to establish that trafficking of K_ATP_ and Kv2.1 by leptin reduces insulin secretion. Ultimately, it will be critical to define the physiological relevance of this mechanism for regulating serum insulin levels and glucose homeostasis in vivo. The few transgenic mouse studies designed to address this question have so far yielded inconsistent results. While early papers support a role of β-cell leptin signaling in glucose regulation [72,73,74], this notion was recently challenged [75]. The cre line used to knock out leptin receptors in β-cells, the mouse strain used, as well as whether animals were subjected to high fat feeding are all factors that could affect experimental outcomes and data interpretation. Thus, these studies that aimed to resolve the longstanding question of the physiological importance of leptin signaling in pancreatic β-cells have instead sparked further controversy. It is evident that more work with careful controls is needed to resolve this important question.

## 7. Concluding Remarks

In conclusion, the trafficking regulation of K_ATP_ channels has emerged as a novel mechanism by which leptin acts peripherally to regulate pancreatic β-cell electrical activity. Although mouse studies have so far not established unequivocally the significance of this regulation in glucose homeostasis, it is encouraging that the limited human β-cell data suggests regulation of channel trafficking by leptin and the underlying signaling mechanism are conserved. The findings that NMDARs mediate the effect of leptin, and are themselves able to regulate β-cell excitability via channel trafficking regulation when activated, are particularly exciting, especially in light of the report by Marquard et al. that demonstrated the inhibition of NMDARs stimulated insulin secretion and improved glucose tolerance in a small type 2 diabetes human trial [43]. Continuing future efforts in delineating downstream leptin signaling mechanisms regulating K_ATP_ channel trafficking in β-cells may offer novel therapeutic targets for improving glycemic control in type 2 diabetes patients by bypassing the leptin-resistance caused by obesity.

## Figures and Tables

**Figure 1 ijms-20-02660-f001:**
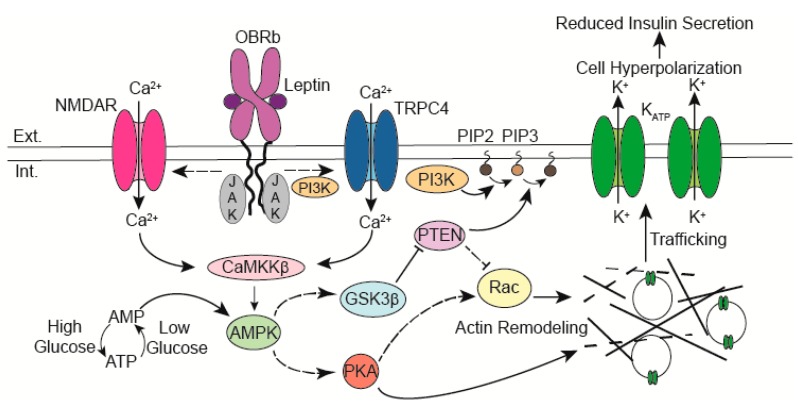
Leptin binding to its receptor (OBRb) leads to a calcium influx through calcium-permeant ion channels. The rise in intracellular calcium activates CaMKKβ which then phosphorylates and activates AMPK. Downstream of AMPK actin remodeling is triggered through PTEN inhibition and activation of PKA. Depolymerization of the cortical actin cytoskeleton allows vesicles containing K_ATP_ channels to traffic to the membrane. The increased abundance of K_ATP_ channels reduces β-cell excitability thereby suppressing insulin secretion. Abbreviations: Extracellular (Ext.), Intracellular (Int.), *N*-methyl-d-Aspartate Receptor (NMDAR), Calcium (Ca^2+^), Janus Kinase (JAK), Ca^2+^/calmodulin kinase kinase β (CaMKKβ), AMP-activated protein kinase (AMPK), Leptin receptor (OBRb), phosphoinositide 3-kinase (PI3K), phosphatidylinositol (4,5)-bisphosphate (PIP2), phosphatidylinositol (3,4,5)-trisphosphate (PIP3), phosphatase and tensin homolog (PTEN), Glycogen synthase kinase 3 β (GSK3β), cAMP-dependent protein kinase A (PKA), Transient receptor potential channel 4 (TRPC4), Potassium (K^+^).

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
