# Peer review of "Leptin-induced Trafficking of KATP Channels: A Mechanism to Regulate Pancreatic β-cell Excitability and Insulin Secretion"

_ijms, 2019, doi:10.3390/ijms20112660_

Round 1
Reviewer 1 Report
This review focuses on comparatively novel mechanism for regulation of insulin secretion in pancreatic beta-cells. The manuscript is summarized about important molecules and excepted signaling pathway for leptin-induced suppression of insulin secretion, and it will be attracted interest from broad research field. Thus, I think this review is suitable for publication on International Journal of Molecular Sciences.
Author Response
None.
Reviewer 2 Report
It is a well written review article summarized recent findings in the regulation of insulin secretion via KATP channel by leptin.
I only have few very minor suggestions:
The format of "KATP" in abstract is not consistent with the rest of the text. The letter "ATP" for "KATP" should be subscripted.
Page 1, line 38-40, the sentence "From these findings it ... for glucose metabolism" is long and hard to understand. Rephrasing this sentence is required. Similar complicated sentences were found throughout this manuscript.
Author Response
Point 1: The format of "KATP" in abstract is not consistent with the rest of the text. The letter "ATP" for "KATP" should be subscripted.
Response 1: "KATP" in the abstract has been corrected.
Point 2: Page 1, line 38-40, the sentence "From these findings it ... for glucose metabolism" is long and hard to understand. Rephrasing this sentence is required. Similar complicated sentences were found throughout this manuscript.
Response 2: This sentence has been simplified and shortened.
Reviewer 3 Report
This review article is well-written and concisely and comprehensively summarized the leptin-induced regulation of KATP channel conductance and trafficking in pancreatic b-cell. I have no specific comment on the manuscript. I would request the authors to cite a paper by Rohit Kulkarni shown below in section 1 or 2. This study indicates that leptin regulates KATP channel-independent effect of GSIS.
Morioka T, et al. Mol Endocrinol 26:967-76, 2012
Author Response
Point 1: I would request the authors to cite a paper by Rohit Kulkarni shown below in section 1 or 2. This study indicates that leptin regulates KATP channel-independent effect of GSIS.
Response 1: Section 1 (lines 76-78) has been modified to inform the reader that there are additional, KATP-independent, mechanisms by which leptin may suppress insulin secretion, but that this review will focus on leptin-mediated KATP channel trafficking.